# Are midwives ready to provide quality evidence-based care after pre-service training? Curricula assessment in four countries—Benin, Malawi, Tanzania, and Uganda

**Ann-Beth Moller**[1]*, **Joanne Welsh**[2], **Elizabeth Ayebare**[3], **Effie Chipeta**[4], **Mechthild M. Gross**[2], **Gisele Houngbo**[5], **Hashim Hounkpatin**[5], **Bianca Kandeya**[4], **Beatrice Mwilike**[6], **Gorrette Nalwadda**[3], **Max Petzold**[1], **Antoinette Sognonvi**[5], **Claudia Hanson**[7,8]

1 School of Public Health and Community Medicine, Institute of Medicine, Sahlgrenska Academy, University of Gothenburg, Gothenburg, Sweden, 2 Midwifery Research and Education Unit, Hannover Medical School, Hannover, Germany, 3 Department of Nursing, Makerere University, Kampala, Uganda, 4 Kamuzu University of Health Sciences, Centre for Reproductive Health, Blantyre, Malawi, 5 Centre de Recherche en Reproduction Humaine et en Démographie (CERRHUD), Cotonou, Benin, 6 School of Nursing, Muhimbili University of Health and Allied Sciences, Dar Es Salaam, Tanzania, 7 Global Public Health, Karolinska Institute, Stockholm, Sweden, 8 Department of Disease Control, London School of Hygiene and Tropical Medicine, London, United Kingdom

* ann-beth.moller.2@gu.se

**Data Availability Statement:** The midwifery pre-service training curricula are not publicly available due to potential confidentiality concerns and only

## Abstract

This research sought to map midwifery pre-service training curricula as part of the Action Leveraging Evidence to Reduce perinatal morTality and morbidity in sub-Saharan Africa (ALERT) project conducted in Benin, Malawi, Tanzania, and Uganda. We conducted the review in two phases. In the first phase, online interviews were performed with the lead project midwives in all four study countries to get an overview of midwifery care providers' pre-service training courses, registration, and licensing requirements. We performed a mapping review of midwifery care providers' pre-service training curricula from different training institutions in the four study countries during the second phase. Curricula were reviewed and mapped against the International Confederation of Midwives (ICM) Essential Competencies framework to assess whether these curricula included the minimum essential training components described in the ICM framework. We identified 10 different professional titles for midwifery care providers. The number of years spent in pre-service training varied from one and a half to four years. Ten pre-service curricula were obtained and the assessment revealed that none of the curricula included all ICM competencies. Main gaps identified in all curricula related to women-centred care, inclusion of women in decision making, provision of care to women with unintended or mistimed pregnancy, fundamental human rights of individuals and evidence-based learning. This review suggests that there are skills, knowledge and behaviour gaps in pre-service training curricula for midwifery care providers when mapped to the ICM Essential Competencies framework. These gaps are similar among the different training courses in participating countries. The review also draws attention to the

shared with the authors for this paper. For access to the curricula the following institutions may be contacted. Benin: Maquette Pédagogique de l'Offre de Formation de Licence Professionnelle en Sciences Infirmières et Obstétricales Option Sage-Femme. Université d'Abomey-Calavi. The contact Université d'Abomey-Calavi, email: vrcireip.uac@uac.bj. Malawi: BSc in Nursing and Midwifery. The Daeyang College of Nursing and Nursing. The Contact Daeyang College, email: Registrar@dyuni.ac.mw. Midwifery Technicians (NMTs). The Christian Health Association of Malawi (CHAM). The contact Pachalo Matchere, email: pmatchere@cham.org.mw. Tanzania: On-line: Level 5. Technician Certificate in Nursing and Midwifery (https://www.stmagdalenenursingschool.com/wp-content/uploads/2019/04/Curriculum-for-NTA-Level-5-Nursing.pdf). Level 6. Ordinary Diploma Nursing and Midwifery (https://www.stmagdalenenursingschool.com/wp-content/uploads/2019/04/Curr_NTA-Level-6.pdf). Revised Competency-based Curriculum for the Bachelor of Science in Nursing (BSCN) Programme. Muhimbili University of Health and Allied Sciences (MUHAS). Competency - Based Curriculum for Bachelor of Science in Midwifery. Muhimbili University of Health and Allied Sciences (MUHAS). The contact person: The Vice Chancellor, Muhimbili University of Health and Allied Sciences, email: vc@muhas.ac.tz. Uganda: BSc of science in midwifery (upgrading) programme. Aga Khan University School of Nursing and Midwifery. The contact person: Grace Edwards, email: grace.edwards@aku.edu. BSs midwifery. Lira University. Contact: Lira University, email: vc@lirauni.ac.ug. Diploma in midwifery. Ministry of Education, Science, Technology and Sports. Contact: Ministry of Education Science, Technology and Sports, email: info@unmeb.go.ug.

**Funding:** This study is part of the ALERT project which is funded by the European Commission's Horizon 2020 (No. 847824) under a call for implementation research for maternal and child health (CH). Publication fees paid by University of Gothenburg, Sweden (ABM). The funders had no role in study design, data collection and analysis, decision to publish, or preparation of the manuscript.

**Competing interests:** The authors have declared that no competing interests exist.

**Abbreviations:** ALERT, Action Leveraging Evidence to Reduce perinatal morTality and morbidity in sub-Saharan Africa project; CDP, Continuing professional development; ENAP, Every Newborn Action Plan; EPMM, Strategies toward ending

plethora of professional titles and different pre-service training curricula within countries.
**Trial registration:** PACTR202006793783148—June 17th, 2020.

## Background

Maternal and newborn health care is key to sustainable development at local and global levels. Despite huge progress in improving health outcomes for women and newborns, many low- and middle-income countries (LMICs), particularly in sub-Saharan Africa, continue to struggle with high rates of maternal and newborn morbidity and mortality [1–5].

A critical progress indicator of the Sustainable Development Goals (SDG) framework is the "proportion of births attended by skilled health personnel" (SDG 3.1.2) [6]. In alignment with the SDG framework, other global initiatives also focus on maternal and newborn health and include the indicator "proportion of births attended by skilled health personnel" in their monitoring frameworks, e.g. the Global Strategy for Women's, Children's and Adolescents' Health, which has described an ambitious action and measurement agenda around the three pillars "Survive, Thrive and Transform" [7] as well as the "Strategies toward ending preventable maternal mortality" (EPMM) [8], and the Every Newborn Action Plan (ENAP) [9]. The competency levels among "skilled health personnel" providing midwifery care and the definition of these providers vary across countries but especially in LMICs. This variation of competence is also revealed when looking at the length and quality of their pre-service training and the many different public and private training programmes [10]. A scoping review identified a total of 102 unique professional titles from 36 LMICs for providers who provide childbirth care and substantial heterogeneity was identified between and within countries related to education, training, as well as core competencies [11]. All the above challenges are also reflected at the global level in the current measurement of "skilled health personnel" providing childbirth care [12].

Evidence suggests that while countries report relatively high levels of birth attendance by skilled health personnel [13,14] maternal and neonatal mortality have not decreased proportionately [15,16].

Pre-service training curricula for midwifery care providers act as a foundation on which students can develop the required competencies to provide quality evidence-based care [17]. The recent 2021 State of the World's Midwifery report called for four key areas of investment for midwives to achieve their potential. One of these key areas is education and training. The report clearly highlights the critical and urgent need for training of midwifery care providers as well as midwifery educators [18]. The World Health Organization Global Strategic Directions for Nursing and Midwifery (2021–2025) put forward four policy priorities key areas of which one is the design of education programmes that are competency-based, that apply effective learning design, that meet quality standards, and that are aligned with population health needs. [19].

To secure further reductions in maternal and newborn mortality, it is imperative to monitor and evaluate to which level and standards midwifery care providers are educated and trained prior to entering the health workforce. At the same time, it is important to ensure that they work in an enabling environment [20]. The International Confederation of Midwives (ICM) Essential Competencies Framework [21] and the ICM Global Standards for Midwifery Education [22] are prominent frameworks that are widely accepted and promoted among 124 countries which have an ICM Association. (https://www.internationalmidwives.org/about-us/membership/).

preventable maternal mortality; ICM, International Confederation of Midwives; LMICs, low-and middle-income countries; SDG, Sustainable Development Goals; UI, Uncertainty interval.

The ICM Essential Competencies Framework outlines the minimum set of knowledge, skills and professional behaviours required by an individual to use the title of a midwife as defined by ICM when entering midwifery practice.

Limited research has been conducted in LMICs to assess the content and quality of curricula for "midwifery care providers" or "skilled health personnel providing maternal and newborn care". This paper presents a review of the evidence as a component of the Action Leveraging Evidence to Reduce perinatal morTality and morbidity in sub-Saharan Africa (ALERT) project—"Positioning Midwifery." ALERT is a hospital maternity-based quality improvement and implementation research project currently implemented in Benin, Malawi, Tanzania and Uganda [23].

The aim of this review was to map pre-service training curricula for midwifery care providers providing antenatal, intrapartum, and postnatal care in the ALERT study countries against the ICM Essential Competencies Framework [21] to inform the development of the ALERT project interventions. It also aims to inform the relevant stakeholders responsible for midwifery care providers' pre-service education in the ALERT study countries about potential gaps in pre-service training curricula.

### Background characteristics of the study countries

Very high maternal mortality ratio is indicated by all four study countries, ranging from 375 (uncertainty interval (UI) 80%: 278–523) in Uganda to 524 (UI 80%: 399–712) in Tanzania [4]. The proportion of births attended by skilled health personnel varies between the countries from 64% in Tanzania to 90% in Malawi, respectively. Stillbirth rates are far behind the 2030 target of 12 or less newborn deaths per 1 000 total births [5], as the 2019 estimates for the four countries range between 16 to 20 stillbirths per 1 000 total births.

Approximately 50% of pregnant women attend four or more antenatal care visits and less than 10% attend eight antenatal visits—just under 2% in Malawi, Tanzania and Uganda compared to 9% in Benin. Wide variation is also observed in the proportion of pregnant women who have their first antenatal care visit within the first four months of pregnancy–ranging from 24% in Malawi and Tanzania, to 32% in Uganda, to 50% in Benin [24] (Table 1).

## Methods

### Ethics statement

This study is part of the Action Leveraging Evidence to Reduce perinatal morTality and morbidity in the sub-Saharan Africa project (ALERT). The project received ethical approval from the following institutions:

Karolinska Institutet, Sweden (Etikprövningsmyndigheten—Dnr 2020–01587).

School of Public Health research and ethics committee (HDREC 808) and Uganda National Council for Science and Technology (UNCST)—(HS1324ES).

Muhimbili University of Health And Allied Sciences (MUHAS) Research and Ethics Committee, Tanzania (MUHAS-REC-04-2020-118) and The Aga Khan University Ethical Review Committee, Tanzania (AKU/2019/044/fb).

College of Medicine Research and Ethics Committee (COMREC), Malawi—(COMREC P.04/20/3038).

Comité National d'Ethique pour la Recherche en Santé, Cotonou, Bénin (83/MS/DC/SGM/CNERS/ST).

The Institutional Review Board at the Institute of Tropical Medicine Antwerp and The Ethics Committee at the University Hospital Antwerp, Belgium—(ITG 1375/20. B3002020000116).

**Table 1. Cross-country comparison of maternal and newborn health indicators.**

| Indicator | Countries | | | |
|---|---|---|---|---|
| | Benin | Malawi | Tanzania | Uganda |
| Proportion of women of reproductive age (aged 15–49 years) who have their need for family planning satisfied with modern methods[1] | 28 | 74 | 55 | 55 |
| Adolescent birth rate (aged 15–19 years) per 1,000 women[1] | 108 | 138 | 139 | 132 |
| Proportion of women aged 15–49 who make their own informed decisions regarding sexual relations, contraceptive use and reproductive health care[1] | 36 | 47 | 47 | 62 |
| Proportion of women of reproductive age 15–49 years who attend antenatal care within the first 4 months of pregnancy[2] | 51 | 24 | 24 | 32 |
| Proportion of women aged 15–49 who received 4 or more antenatal care visits[2] | 52 | 51 | 62 | 57 |
| Proportion of women aged 15–49 who received 8 or more antenatal care visits[2] | 9 | 1.4 | 1.3 | 1.2 |
| Proportion of births attended by skilled health personnel (aged 15–49)[1] | 78 | 90 | 64 | 74 |
| Proportion of births in a health facility[2] | 84 | 91 | 63 | 73 |
| Proportion of births by caesarean section[2] | 5 | 6 | 6 | 6 |
| Proportion of newborns with skin-to-skin contact immediately after birth[2] | 75 | 65 | 29 | 73 |
| Proportion of newborns who started breastfeeding within one hour of birth[2] | 54 | 76 | 51 | 66 |
| Proportion of women who have postpartum contact with a health provider within 2 days of delivery (aged 15–49)[2] | 7 | 9 | 3 | 3 |
| Proportion of newborns who have postnatal contact with a health provider within 2 days of delivery[2] | 7.8 | 12 | 4 | 3 |
| Maternal mortality ratio per 100 000 live births (2017)[1] | 397 (UI*: 291–570) | 349 (UI: 244–507) | 524 (UI: 399–712) | 375 (UI: 278–523) |
| Neonatal mortality rate per 1 000 live birth (2019)[1] | 31 | 20 | 20 | 20 |
| Stillbirth rate per 1 000 total births (2019)[3] | 20 (UI**: 17–24) | 16 (UI: 14–18) | 18 (UI: 14–25) | 18 (UI: 16–19) |
| Preterm birth rate per 100 live births (2014)[4] | 9.3 | 10.5 | 16.6 | 9.0 |

*Uncertainty interval (UI) 80%.

** Uncertainty interval (UI) 90%.

[1] Global Sustainable Development Goals Indicators Database. Downloaded February 22, 2022 (latest available data included). Available at: https://unstats.un.org/sdgs/indicators/database/.

[2] ICF. The DHS Program STATcompiler. Funded by USAID. http://www.statcompiler.com (latest available data included). Downloaded February 22, 2022.

[3] United Nations Inter-Agency Group for Child Mortality Estimation (UN IGME), 'A Neglected Tragedy: The global burden of stillbirths', United Nations Children's Fund, New York, 2020.

[4] Chawanpaiboon S, Vogel JP, Moller AB et al. Global, regional, and national estimates of levels of preterm birth in 2014: A systematic review and modelling analysis. Lancet Glob Health. 2019 Jan;7(1):e37-e46.

## Study design and settings

A mapping review [25] was considered to be the most appropriate research method to map out the pre-service training curricula for midwifery care providers in the four ALERT study

countries against the ICM Essential Competencies Framework. No formal quality assessment appraisal was acquired for this type of review.

In addition to the mapping review, online interviews with the lead ALERT study midwives were conducted to elucidate additional information concerning the education system and national registration and licensing requirements of midwifery care providers in the study countries.

## Data collection

Data collection was carried out in two phases during 2021–2022. In the first phase, ABM and JW conducted online interviews in English with the four lead country midwives in the ALERT project. The lead midwives were purposely selected as they were considered well informed and up-to-date to provide a general overview of the pre-service education system and the professional titles of midwifery care providers, as well as the local registration and licencing processes and professional organizations in each study country. All the lead midwives provide pre-service training, are registered and licensed to work in their capacity and part of the professional organizations (S1 Text). The second phase of data collection included the mapping review of the pre-service training curricula for midwifery care providers providing antenatal, intrapartum, and postnatal care to the ICM Essential Competencies for Midwifery Practice [21]. To obtain curricula we first conducted an on-line search and only located curricula for Tanzania. A curricula request letter was sent to the four ALERT project PIs training institutions, the national nursing and midwifery associations, and the World Health Organization's country offices in the four countries addressed to the focal points for nursing and midwifery. The curricula included in this mapping are curricula which we were able to obtain after several follow-ups and personal contacts.

## Data analysis

Data from the structured online interviews were entered into Microsoft Excel at the time of interview and summarized. An extraction form in Microsoft Excel was developed to map the pre-service training curricula to the ICM Framework [21]. As summarized in Box 1. the ICM framework consists of four main categories: i) general competencies, ii) pre-pregnancy and antenatal care, iii) care during labour and birth, and iv) ongoing care of women and newborns. Within each of these four categories indicators related to knowledge and skills and behaviours are outlined. Overall, the ICM framework includes 317 indicators– 132 relate to knowledge and 185 to skills and behaviours. Data extraction was performed by the two first authors (ABM and JW) and checked by the country lead midwives (S1 Table). Each curriculum was thoroughly read several times and reviewed for content to determine if the indicators in the ICM framework were included in the curriculum. Indicators were considered as equally important–i.e. one point for each indicator—and hence the maximum score possible for a curriculum was 317 which denotes 100% alignment with the ICM framework. A curriculum was deemed to have met the essential competencies if it included similar objectives and content as the ICM framework. For each curricula summary scores and percentages (%) were calculated for knowledge components and skills/behaviours components within each competency category. In addition, summary scores for each of the four main categories as well as total scores for each curricula were calculated.

## Ethical consideration

This study is part the Action Leveraging Evidence to Reduce perinatal morTality and morbidity in sub-Saharan Africa (ALERT) project. The project received ethical approval from the local and national institutional review boards as follows:

## Box 1. ICM framework–competency categories and indicators [21]

| Competency categories and indicators | No. of indicators per competency in ICM framework | |
|---|---|---|
| **1. General competencies t** | | |
| 1.a Assume responsibility for own decisions and actions as an autonomous practitioner | Knowledge | 5 |
| | Skills and behaviours | 4 |
| 1.b Assume responsibility for self-care and self-development as a midwife | Knowledge | 1 |
| | Skills and behaviours | 6 |
| 1.c Appropriately delegate aspects of care and provide supervision | Knowledge | 3 |
| | Skills and behaviours | 2 |
| 1.d Use research to inform practice | Knowledge | 3 |
| | Skills and behaviours | 2 |
| 1.e Uphold fundamental human rights of individuals when providing midwifery care | Knowledge | 4 |
| | Skills and behaviours | 5 |
| 1.f Adhere to jurisdictional laws, regulatory requirements, and codes of conduct for midwifery practice | Knowledge | 4 |
| | Skills and behaviours | 7 |
| 1.g Facilitate women to make individual choices about care | Knowledge | 3 |
| | Skills and behaviours | 4 |
| 1.h Demonstrate effective interpersonal communication with women and families, health care teams, and community groups | Knowledge | 5 |
| | Skills and behaviours | 10 |
| 1.i Facilitate normal birth processes in institutional and community settings, including women's homes | Knowledge | 5 |
| | Skills and behaviours | 3 |
| 1.j Assess the health status, screen for health risks, and promote general health and well-being of women and infants | Knowledge | 3 |
| | Skills and behaviours | 6 |
| 1.k Prevent and treat common health problems related to reproduction and early life | Knowledge | 4 |
| | Skills and behaviours | 6 |
| 1.l Recognise abnormalities and complications and institute appropriate treatment and referral | Knowledge | 5 |
| | Skills and behaviours | 7 |
| 1.m Care for women who experience physical and sexual violence and abuse | Knowledge | 3 |
| | Skills and behaviours | 6 |
| *1. Sub-total* | *Knowledge* | *48* |

(*Continued*)

**Box 1**. (Continued)

| | | |
|---|---|---|
| | *Skills and behaviours* | *68* |
| **2. Competencies specific to pre- pregnancy and antenatal care** | | |
| 2.a Provide pre-pregnancy care | Knowledge | 3 |
| | Skills and behaviours | 4 |
| 2.b Determine health status of woman | Knowledge | 4 |
| | Skills and behaviours | 7 |
| 2.c Assess fetal well-being | Knowledge | 2 |
| | Skills and behaviours | 3 |
| 2.d Monitor the progression of pregnancy | Knowledge | 4 |
| | Skills and behaviours | 5 |
| 2.e Promote and support health behaviours that improve wellbeing | Knowledge | 7 |
| | Skills and behaviours | 5 |
| 2.f Provide anticipatory guidance related to pregnancy, birth, breastfeeding, parenthood, and change in the family | Knowledge | 3 |
| | Skills and behaviours | 5 |
| 2.g Detect, stabilise, manage, and refer women with complicated pregnancies | Knowledge | 4 |
| | Skills and behaviours | 5 |
| 2.h Assist the woman and her family to plan for an appropriate place of birth | Knowledge | 3 |
| | Skills and behaviours | 3 |
| 2.i Provide care to women with unintended or mistimed pregnancy | Knowledge | 7 |
| | Skills and behaviours | 8 |
| ADDITIONAL SKILL: An additional skill is performed by midwives under either of two circumstances: a) Midwives who elect to engage in a broader scope of practice and/or b) Midwives who have to implement certain skills to make a difference in maternal and neonatal outcome. | | 2 |
| *2. Sub-total* | *Knowledge* | *37* |
| | *Skills and behaviours* | *47* |
| ***3. Competencies specific to care during labour and birth*** | | |
| 3.a Promote physiologic labour and birth | Knowledge | 6 |
| | Skills and behaviours | 14 |
| 3.b Manage a safe spontaneous vaginal birth; prevent, detect and stabilise complications | Knowledge | 6 |
| | Skills and behaviours | 14 |
| 3.c Provide care of the newborn immediately after birth | Knowledge | 7 |
| | Skills and behaviours | 7 |
| *3. Sub-total* | *Knowledge* | *19* |
| | *Skills and behaviours* | *35* |
| ***4. Competencies specific to the ongoing care of women and newborns*** | | |
| 4.a Provide postnatal care for the healthy woman | Knowledge | 4 |

(*Continued*)

**Box 1**. (Continued)

| | | |
|---|---|---|
| | Skills and behaviours | 6 |
| 4.b Provide care to healthy newborn infant | Knowledge | 5 |
| | Skills and behaviours | 4 |
| 4.c Promote and support breastfeeding | Knowledge | 6 |
| | Skills and behaviours | 7 |
| 4.d Detect, treat, and stabilise postnatal complications in woman and refer as necessary | Knowledge | 3 |
| | Skills and behaviours | 6 |
| 4.e Detect stabilise, and manage health problems in newborn infant and refer if necessary | Knowledge | 5 |
| | Skills and behaviours | 6 |
| 4.f Provide family planning services | Knowledge | 5 |
| | Skills and behaviours | 6 |
| *4. Sub-total* | *Knowledge* | *28* |
| | *Skills and behaviours* | *35* |
| **Total knowledge** | | **132** |
| **Total skills and behaviours** | | **185** |
| **TOTAL INDICATORS** | | **317** |

Uganda National Council for Science and Technology.

Muhimbili University of Health and Allied Sciences.

Research and Ethics Committee, Tanzania and The Aga Khan University Ethical Review Committee, Tanzania.

College of Medicine. Research and Ethics Committee, Malawi.

Comité National d'Ethique pour la Recherche en Santé, Cotonou, Bénin.

The Institutional Review Board at the Institute of Tropical Medicine Antwerp and the Ethics Committee at the University Hospital Antwerp, Belgium.

All participants provided verbal consent to participate in the study.

## Results

### Professional titles and years of training

The four online interviews with lead country midwives in each of the study countries, elucidated 10 different midwifery care provider professional titles among providers working in the maternity wards. Table 2. presents these professional titles, training programme, duration and academic award. Benin was the only country with just two professional titles for midwifery, compared to four in Uganda, and five in Malawi and Tanzania. Benin is also unique in the sense that midwifery is a bachelor level qualification only.

Supplementary information (S2 Table) includes additional information about the educational systems, entry requirements, and details for obtaining and maintaining the professional titles. The number of years spent in pre-service training varied across countries from one and a half to four years. Registration and licensing to maintain practice also varied across the countries. In Benin the authorization to work in the public sector is provided by the Ministry of

**Table 2. Professional titles and training programme.**

| Country | Professional title | Training programme | Duration | Award |
|---|---|---|---|---|
| Benin | Registered Midwife | Bachelor in Midwifery | 3 years | Bachelor |
| | Registered Nurse | Bachelor in Nursing | 3 years | Bachelor |
| Malawi | Enrolled Nurse- Midwife | Certificate in Nursing and Midwifery | 2 Years | Certificate |
| | Nurse-Midwife Technician | Diploma in Nursing and Midwifery | 3 years | Diploma |
| | Registered Nurse-Midwife | Diploma in Nursing and Certificate in Midwifery | 4 years | 3 years Diploma in Nursing and 1 year Certificate in Midwifery |
| | Registered Nurse- Midwife | Bachelor in Midwifery | 4 years | Bachelor |
| | Advanced Midwife Practitioner | Master of Science inMidwifery | 4 years | Master |
| Tanzania | Enrolled Nurse-Midwife | Certificate in Nursing and Midwifery | 2 years | Certificate |
| | Registered Nurse-Midwife | Diploma in Nursing and Midwifery | 3 years (and some have been trained for 4 years) | Diploma |
| | Registered Midwife | Bachelor of Science in Midwifery | 4 years | Bachelor |
| | Registered Nurse | Bachelor of Science in Nursing | 4 years | Bachelor |
| | Master of Science in Midwifery and Women's Health | Master of Science in Midwifery and Women's Health | 2 Years | Master |
| Uganda | Enrolled Midwife | Certificate in Midwifery | 2.5 years | Certificate |
| | Registered Midwife | Diploma in Midwifery (Upgrading, Top up) | 18 months | Diploma |
| | Registered Midwife | Diploma in Midwifery | 3 years | Diploma |
| | Registered Bachelor of Science in Midwifery | Bachelor of Science in Midwifery | 4 years | Bachelor |
| | Masters in Nursing/Midwifery | Master of Nursing (Midwifery and Women's Health) | 2 years | Master |

Health when pre-service training is completed and authorization to work for private clients is likewise provided by a Ministry of Health and is currently a lifetime license. In Malawi all licences are renewed yearly through the Nursing and Midwifery Council. In the case of Tanzania licence renewal is needed every three years and is subject to midwifery care providers' continuing professional development (CPD) points. In Uganda renewal of licence is required every three years and CPD points are also required but currently the regulation is not strongly implemented. Entry requirements to pre-service training differed between countries. However, due to different schooling systems in each country, a meaningful comparison of entry requirements was not feasible.

## Pre-service training competencies

The sharing of curricula is not commonplace in many countries as these are considered confidential documents within training institutions which has also been the case to a certain extent in our study countries. However, we have been fortunate to obtain pre-service curricula from ten institutions: one from Benin, two from Malawi, four from Tanzania and three from Uganda. Due to restrictions of sharing curricula, which are seen as confidential documents, we could not obtain curricula for all professional titles that provide midwifery care in each country.

**Table 3. Summary scores for the four categories and total scores of indicators in country curriculum mapped to the ICM framework and years of training programmes.**

| Competency categories | No. of indicators per competency in the ICM framework | Benin (3 yrs) | | Malawi (4 yrs) | | Malawi (3 yrs) | | Tanzania (3 yrs) | | Tanzania (2 yrs) | | Tanzania (4 yrs) | | Tanzania (4 yrs) | | Uganda (2.5 yrs) | | Uganda (4 yrs) | | Uganda (3 yrs) | |
|---|---|---|---|---|---|---|---|---|---|---|---|---|---|---|---|---|---|---|---|---|---|
| | | N | % | N | % | N | % | N | % | N | % | N | % | N | % | N | % | N | % | N | % |
| *1. GENERAL COMPETENCIES* | | | | | | | | | | | | | | | | | | | | | |
| 1. Sub-total | Knowledge | 48 | 30 | 63 | 38 | 79 | 35 | 73 | 36 | 75 | 30 | 63 | 32 | 67 | 35 | 73 | 37 | 77 | 33 | 69 | 27 | 56 |
| | Skills and behaviours | 68 | 24 | 35 | 42 | 62 | 31 | 46 | 34 | 50 | 27 | 40 | 38 | 56 | 46 | 68 | 46 | 68 | 39 | 57 | 38 | 56 |
| **1. Total** | | 116 | 54 | 47 | 80 | 69 | 66 | 57 | 70 | 60 | 57 | 49 | 70 | 60 | 81 | 70 | 83 | 72 | 72 | 62 | 65 | 56 |
| *2. COMPETENCIES SPECIFIC TO PRE- PREGNANCY AND ANTENATAL CARE* | | | | | | | | | | | | | | | | | | | | | |
| 2. Sub-total | Knowledge | 37 | 12 | 32 | 20 | 54 | 17 | 46 | 21 | 57 | 21 | 57 | 21 | 57 | 24 | 65 | 28 | 76 | 20 | 54 | 19 | 51 |
| | Skills and behaviours | 47 | 23 | 49 | 27 | 57 | 21 | 45 | 26 | 55 | 26 | 55 | 19 | 40 | 30 | 64 | 33 | 70 | 21 | 45 | 26 | 55 |
| **2. Total** | | 84 | 35 | 42 | 47 | 56 | 38 | 45 | 47 | 56 | 47 | 56 | 40 | 48 | 54 | 64 | 61 | 73 | 41 | 49 | 45 | 54 |
| *3. COMPETENCIES SPECIFIC TO CARE DURING LABOUR AND BIRTH* | | | | | | | | | | | | | | | | | | | | | |
| 3. Sub-total | Knowledge | 19 | 14 | 74 | 17 | 89 | 18 | 95 | 18 | 95 | 17 | 89 | 14 | 74 | 18 | 95 | 18 | 95 | 11 | 58 | 17 | 89 |
| | Skills and behaviours | 35 | 18 | 51 | 25 | 71 | 23 | 66 | 25 | 71 | 22 | 63 | 16 | 46 | 27 | 77 | 31 | 89 | 17 | 49 | 21 | 60 |
| **3. Total** | | 54 | 32 | 59 | 42 | 78 | 41 | 76 | 43 | 80 | 39 | 72 | 30 | 56 | 45 | 83 | 49 | 91 | 28 | 52 | 38 | 70 |
| *4. COMPETENCIES SPECIFIC TO THE ONGOING CARE OF WOMEN AND NEWBORNS* | | | | | | | | | | | | | | | | | | | | | |
| 4. Sub-total | Knowledge | 28 | 14 | 50 | 22 | 79 | 17 | 61 | 16 | 57 | 12 | 43 | 16 | 57 | 18 | 64 | 19 | 68 | 20 | 71 | 16 | 57 |
| | Skills and behaviours | 35 | 13 | 37 | 29 | 83 | 21 | 60 | 23 | 66 | 17 | 49 | 17 | 49 | 24 | 69 | 30 | 86 | 19 | 54 | 20 | 57 |
| **4. Total** | | 63 | 27 | 43 | 51 | 81 | 38 | 60 | 39 | 62 | 29 | 46 | 33 | 52 | 42 | 67 | 49 | 78 | 39 | 62 | 36 | 57 |
| **Total knowledge** | | 132 | 70 | 53 | 97 | 73 | 87 | 66 | 91 | 69 | 80 | 61 | 83 | 63 | 95 | 72 | 102 | 77 | 84 | 64 | 79 | 60 |
| **Total skills and behaviours** | | 185 | 78 | 42 | 123 | 66 | 96 | 52 | 108 | 58 | 92 | 50 | 90 | 49 | 127 | 69 | 140 | 76 | 96 | 52 | 105 | 57 |
| **TOTAL all indicators** | | 317 | 148 | 47 | 220 | 69 | 183 | 58 | 199 | 63 | 172 | 54 | 173 | 55 | 222 | 70 | 242 | 76 | 180 | 57 | 184 | 58 |

The ten curricula were published between 2014 and 2020 and the pre-service training varied from two to four years. The number of hours within the training programmes varied from 4,199 to 5,430 hours for the Diploma programmes and from 5,264 to 5,713 hours for the Bachelor programmes. The ICM recommends that the minimum length of a direct-entry midwifery education programme should be 36 months which corresponds to approximately 4,600 hours acknowledging that this varies from region to region [26].

Full details about the scoring for each of the pre-service curriculum as well as the years of training can be found in S3 Table and Table 3. provides a summary of the four main categories and total scores of all indicators. We found that the percentage score of all ICM Essential Competencies for Midwifery practice included in the curricula ranged from 47% in Benin to 76% in Uganda (midwifery upgrading course), respectively. For the four main categories: i) general competencies, ii) pre-pregnancy and antenatal care, iii) care during labour and birth, and iv) ongoing care of women and newborns, none of the curricula comprehensively included all the underlying competencies.

The best scores for all the curricula were in the third category "care during labour and birth" with the maximum score of 91% and no scores lower than 59% for all the curricula. The lowest scores within the categories were in the second category "pre-pregnancy and antenatal care" such as "promote and support health behaviours that improve wellbeing", "assist the women and her family to plan an appropriate place of birth", and "provide care to women with unintended or mistimed pregnancy". A mixed picture emerged for the first main category with several indicators with relatively high scores and other indicators not included in many curricula.

The low scores indicators apply in particular to the indicators; "facilitate women to make individual choices about care", "appropriately delegate aspects of care and provide supervision", "reference to national and/or international guidelines and evidence to inform best practice, "care for women who experience physical and sexual violence and abuse" and "uphold fundamental human rights of individuals when providing midwifery care".

From the review it was also observed that none of the curricula indicated information about plans and timelines for reviewing and updating content, and only two curricula acknowledge that the ICM competency framework was considered in the development of the curriculum.

In addition, the course literature reading lists were mainly based on outdated textbooks and there was little evidence of reference literature to national and international guidelines or important scientific papers.

## Discussion

To the best of our knowledge, this is the first review to comprehensively map pre-service training curricula for midwifery care providers providing pregnancy, childbirth, and postpartum care in Benin, Malawi, Tanzania and Uganda against the ICM Essential Competencies Framework [21].

The review identified gaps in terms of knowledge and skills being taught, which may result in poor competence, thereby affecting the evidence-based quality care midwifery care providers are able to provide once qualified and entering the health workforce. Our findings indicate that gaps in pre-service training curricula were consistent across the study countries, with a lack of focus on woman centered care, information sharing and shared decision making, care related to women who experience physical and sexual violence and abuse as well as aspects concerning fundamental human rights when providing midwifery care. These aspects of respectful care have been described as missing in a number of studies despite being seen as very critical in relation to women's experiences during the pregnancy, childbirth and postnatal continuum, as well as important in promoting the use of facility-based maternity care [27–31].

The curricula also reflect the legal status of abortion in the study countries [32] and competencies related to the provision of care to women with unintended or mistimed pregnancy were only partly included in five pre-service training curricula. The "additional skill" (see Box 1) in the second category was not included in any of the curricula.

We are aware that in the study countries both private and public institutions co-exist and wondered about differences in the quality of pre-service training among these institutions. As we were unable to review curricula for both public and private institutions, we cannot say if standards differ between type of institution. A review done in the African region during 2018 revealed that two, nine, 84 and 71 nursing and midwifery institutions exist in Benin, Malawi, Tanzania and Uganda respectively but no sector details are provided for these countries [33].

Assessing curricula is one track followed to measure quality of pre-service education but does not capture key information on the quality of training provided, the educational content delivered in the classroom nor in clinical practice. A recent assessment carried out in Tanzania [34] found that curricula were not implemented as intended as nursing and midwifery programme educators lacked understanding of the competency-based curriculum. In addition, educators reported that lack of time and equipment coupled with large class sizes impeded their ability to facilitate participatory teaching methods [34]. A meta-synthesis focusing on curriculum reform to competency-based curricula in the African context identified challenges such as outdated and poorly designed curricula, lack of involvement from students and

educators in curricula design, and educators lacking pedagogical skills to adequately implement competency-based curricula [35].

Whilst there is a lack of research investigating the competence levels of graduating midwifery students in our study countries, evidence from Ethiopia indicates inadequate levels of competence among graduating midwifery students. The authors concluded that in-service training is required for newly graduated midwifery students as soon as they enter the profession [36]. The recently published Regional Midwifery report for East and Southern Africa concluded that in many of the countries in the region the quality of pre-service education and training do not meet global standards due to, in many instances, outdated curricula, faculty assessed as being inadequately equipped, inadequate clinic-based practice during training as well as continuing professional development. The report also highlights the need for strengthening of education accreditation systems, regular curricula reviews to ensure alignment with new evidence and recommendations for more focus on quality of care alongside respectful care [10].

Meeting the EPMM [37] and the ENAP targets by 2025 [38] as well as the SDG targets by 2030 [6], will require accelerated efforts to ensure reductions in stillbirth, and maternal and newborn mortality and morbidity [39].

Ensuring midwifery care providers level-up their training to international and/or national standards after their pre-service education needs specific interventions and broader evaluation. Curriculum assessment will play a critical part as a first step to identify competence gaps, but should not be the only component when evaluating standards of education [22,26]. Other components such as the programme governance, teaching capacity in both theory and clinical practice, programme resources and the physical teaching environment, teacher's qualification, entry requirements, pedagogical methods and assessment strategies, quality improvement and others are all factors which interact and should be considered to ensure well trained midwifery care providers who can deliver positive maternity experiences and contribute to decreases in maternal and newborn morbidity and mortality.

The strengths of this review lie in the first-time use of a comprehensive and systematic mapping of the included curricula using the four ICM Essential Competencies categories [21], which include 317 knowledge, and skills and behaviours indicators. The limitations of the mapping review include our inability to review curricula for all healthcare professionals who provide midwifery care in the study countries. Additionally, the study has not comprehensively covered evidence/curricula from all types of institutions providing midwifery training and education (e.g., private for profit and private non-profit institutions/faculties). The majority of the curricula were published and implemented before the ICM competency framework 2019 was released and consequently the assessment may to some extent not reflect the current available recommendation at the time of drafting the curricula. However the review revealed that only two curricula acknowledged the used of the ICM competency framework when developing the curriculum hence the previous ICM global standards for midwifery education framework (2013) [40] may not have impacted and influenced the content. A study published in 2019 confirm the limited use and implementation of ICM education standards to ensure quality midwifery education [41]. As all of these curricula are currently in use, we see value in using the 2019 ICM competency framework to assess gaps that could be taken into consideration when planning in-service training and at the time of curriculum revision.

The mapping of the curricula provides theoretical knowledge of the intended learning outcomes but does not provide information about the educators' academic qualifications, the teaching environment, access to evidence for best care, training equipment, and clinical practice.

There are also other important aspects that this study did not cover such as the regulation of training programmes, the assessment of training institutions and accreditation. However, a recent study highlighted the need for better global, regional and national alignment with professional regulation and practice as the current systems have hindered service delivery and progress towards achieving universal health coverage [33]. Midwife trainees and pre-service training educators perspectives on the curricula content, usefulness and implementation were not assessed in this review, and would be important views to further illuminate, enhance and elaborate on our review findings.

This study also does not enable an analysis of student supervision and mentorship during both theoretical teaching and clinical practice placements which are essential components of pre-service training and more insight into these aspects is warranted.

Several gaps in the curricula were identified and countries are urged to try to address these aspects. However, positive aspects of the curricula should also be stressed. Many institutions have included many pertinent stakeholders in the development and design of the curricula, and should be encouraged to continue to update as well as develop the content according to local health workforce needs and the education system expansion.

## Conclusion and implications for policy

Despite limitations, this study provides important policy and programme planning measures that could be the first step to a more comprehensive assessment of factors that impact how prepared midwifery care providers are once qualified and starting their first job. Improved pre-service education in many training institutions is needed to ensure that midwifery care providers are competent to provide quality evidence-based maternal and newborn care. When developing, designing or updating curricula training institutions and stakeholders could use the ICM Essential Competencies as the guiding framework as basic knowledge, skills and behaviours which are critical in all settings are integrated into the framework. To further enhance the curricula it would be critical to ensure that latest evidence based guidelines related to maternal and newborn care and literature are reflected and accessible to midwifery students as well as educators. However, to better inform national and local policies and programmes, additional information is needed to understand the differences between what is included in the curricula and what is taught in the classroom and in clinical practice.

Additional research would also support a better understanding of the academic level and competencies of faculty educators, the learning environment, the access to new evidence and skill labs, clinical practice as well as supervision and mentorship.

The World Health Organization designated 2020 as the International Year of the Nurse and the Midwife [42] and 2021 as the International Year of Health and Care Workers [43]. These high-visibility campaigns will hopefully inspire future action and research into effective interventions and strategies that target midwifery pre-service education to improve maternal and newborn care and outcomes.

## Supporting information

**S1 Table. Data extraction form.**
(PDF)

**S2 Table. Education programme details.**
(PDF)

**S3 Table. Number of indicators in country curriculum.**
(PDF)

**S1 Text. Interview guide.**
(PDF)

## Acknowledgments

We would like to thank the training institutions which kindly shared curriculum for this review.

## Author Contributions

**Conceptualization:** Ann-Beth Moller.

**Data curation:** Ann-Beth Moller, Joanne Welsh, Elizabeth Ayebare, Bianca Kandeya, Beatrice Mwilike.

**Formal analysis:** Ann-Beth Moller, Joanne Welsh.

**Funding acquisition:** Max Petzold, Claudia Hanson.

**Methodology:** Ann-Beth Moller, Joanne Welsh.

**Project administration:** Claudia Hanson.

**Supervision:** Max Petzold, Claudia Hanson.

**Writing – original draft:** Ann-Beth Moller.

**Writing – review & editing:** Ann-Beth Moller, Joanne Welsh, Elizabeth Ayebare, Effie Chipeta, Mechthild M. Gross, Gisele Houngbo, Hashim Hounkpatin, Bianca Kandeya, Beatrice Mwilike, Gorrette Nalwadda, Max Petzold, Antoinette Sognonvi, Claudia Hanson.

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
