## [Decision Letter · Decision Letter 0]

21 Jul 2022

PGPH-D-22-00810

Are midwives ready to provide quality evidence-based care after pre-service training? Curricula assessment in four countries - Benin, Malawi, Tanzania, and Uganda

Dear Dr. Moller,

Thank you for submitting your manuscript to PLOS Global Public Health. After careful consideration, we feel that it has merit but does not fully meet PLOS Global Public Health’s publication criteria as it currently stands. Therefore, we invite you to submit a revised version of the manuscript that addresses the points raised during the review process.

We look forward to receiving your revised manuscript.

Kind regards,

Hannah Tappis, DrPH, MPH

Academic Editor

Journal Requirements:

1. Please amend your detailed online Financial Disclosure statement. This is published with the article. It must therefore be completed in full sentences and contain the exact wording you wish to be published.

2. Please ensure that the funders and grant numbers match between the Financial Disclosure field and the Funding Information tab in your submission form. Note that the funders must be provided in the same order in both places as well.

3. Please update your online Competing Interests statement. If you have no competing interests to declare, please state: “The authors have declared that no competing interests exist.”

4. Please provide additional details regarding participant consent. In the ethics statement in the Methods section, please ensure that you have also specified whether consent was informed.

5. We have noticed that you have uploaded Supporting Information files, but you have not included a list of legends. Please add a full list of legends for your Supporting Information files after the references list.

Additional Editor Comments (if provided):

Reviewers' comments:

Reviewer's Responses to Questions

**Comments to the Author**

1. Does this manuscript meet PLOS Global Public Health’s publication criteria? Is the manuscript technically sound, and do the data support the conclusions? The manuscript must describe methodologically and ethically rigorous research with conclusions that are appropriately drawn based on the data presented.

Reviewer #1: Yes

Reviewer #2: Yes

2. Has the statistical analysis been performed appropriately and rigorously?

Reviewer #1: Yes

Reviewer #2: N/A

3. Have the authors made all data underlying the findings in their manuscript fully available (please refer to the Data Availability Statement at the start of the manuscript PDF file)?

Reviewer #1: Yes

Reviewer #2: Yes

4. Is the manuscript presented in an intelligible fashion and written in standard English?

Reviewer #1: Yes

Reviewer #2: Yes

5. Review Comments to the Author

Reviewer #1: The authors have used a mapping exercise to identify evidence-based care after services training for midwives across 4 countries, Benin, Malawi, Tanzania, and Uganda. The methodology is sound, and the manuscript is well-written. The article is acceptable in the current form.

Reviewer #2: In this manuscript authors have reported on the findings of the mapping of the curricula for the training of midwives in four African countries. The manuscript makes important contributions to the literature and identify gaps that should be worked on to improve the curricula.

However, I have a few observations I will like to make known to the authors, that I believe will help improve the readability of the manuscript.

1. The abstract and the background are well written and have situated the topic in a relevant context. The problem statement is well articulated and the significance of the study is well defined

2. The methods are well written except that there is lack of clarity on how authors requested for the midwifery curricula of the various schools. How was the request done? Was it sent to all training schools in the respective countries or they were selected? If they were selected, was it a random process or non-random?

3. Another issue that appears unclear to me has to do with how authors determined if a curriculum met the ICM framework? Was it that a curriculum was deemed to have met the ICM framework if they had objectives or content that are similar with the ICM?

4. In the results section authors indicated that they conducted four interviews with lead country midwives. In my opinion, four seems too small to get rich information for this kind of review. Authors could have invited midwives from other schools within the project countries that were not part of the project.

5. In the discussion section, it is commendable that authors noted that assessment of the curricula is an important necessary step but it is not enough to bring about improvement in quality of training given that there are several facets that should be considered in the assessment process. One of such facets which was not noted by the authors, is the beneficiaries of the curriculum i.e., midwifery trainees. Their perspectives will have illuminated further the findings presented in this study if authors had also interviewed them. This should be noted as limitation of the study.

6. It is also worthwhile for authors to discuss at least one positive finding in the current curriculum that they will encourage the institutions to maintain in the curriculum. Notwithstanding the fact that the review was intended to identify gaps it will enrich the report if they also report on the positive aspects of the curriculum.

7. Authors should also note that health curricula are designed to meet the local health needs of the country or community. I am wondering if they reflected their findings in relation to this key component of health curricula design.

8. Authors should also offer a few solutions or recommendations that could be adopted to help improve the curricula.

6. PLOS authors have the option to publish the peer review history of their article (what does this mean?). If published, this will include your full peer review and any attached files.

**Do you want your identity to be public for this peer review?** For information about this choice, including consent withdrawal, please see our Privacy Policy.

Reviewer #1: **Yes: **Denny John

Reviewer #2: **Yes: **A/Prof. Victor Mogre

---

## [Editor Report · Decision Letter 1]

26 Aug 2022

Are midwives ready to provide quality evidence-based care after pre-service training? Curricula assessment in four countries - Benin, Malawi, Tanzania, and Uganda

PGPH-D-22-00810R1

Dear PhD student Moller,

We are pleased to inform you that your manuscript 'Are midwives ready to provide quality evidence-based care after pre-service training? Curricula assessment in four countries - Benin, Malawi, Tanzania, and Uganda' has been provisionally accepted for publication in PLOS Global Public Health.

Best regards,

Hannah Tappis, DrPH, MPH

Academic Editor